# Two-dimensional flow nanometry of biological nanoparticles for accurate determination of their size and emission intensity

Stephan Block[1,†], Björn Johansson Fast[1], Anders Lundgren[1,2], Vladimir P. Zhdanov[1,3] & Fredrik Höök[1]

Biological nanoparticles (BNPs) are of high interest due to their key role in various biological processes and use as biomarkers. BNP size and composition are decisive for their functions, but simultaneous determination of both properties with high accuracy remains challenging. Optical microscopy allows precise determination of fluorescence/scattering intensity, but not the size of individual BNPs. The latter is better determined by tracking their random motion in bulk, but the limited illumination volume for tracking this motion impedes reliable intensity determination. Here, we show that by attaching BNPs to a supported lipid bilayer, subjecting them to hydrodynamic flows and tracking their motion via surface-sensitive optical imaging enable determination of their diffusion coefficients and flow-induced drifts, from which accurate quantification of both BNP size and emission intensity can be made. For vesicles, the accuracy of this approach is demonstrated by resolving the expected radius-squared dependence of their fluorescence intensity for radii down to 15 nm.

[1] Department of Physics, Division of Biological Physics, Chalmers University of Technology, Gothenburg SE-412 96, Sweden. [2] Department of Nanobiotechnology, University of Natural Resources and Life Sciences, Vienna 1190, Austria. [3] Boreskov Institute of Catalysis, Russian Academy of Sciences, Novosibirsk 630090, Russia. † Present address: Department of Chemistry and Biochemistry, Freie Universität Berlin, 14195 Berlin, Germany. Correspondence and requests for materials should be addressed to S.B. (email: stephan.block@fu-berlin.de) or to F.H. (email: fredrik.hook@chalmers.se).

**B**iological nanoparticles (BNPs) like viruses, micelles, exosomes, vesicles or biologically functionalized particles are of high relevance for current research as they are involved in a multitude of processes (for example, transmission of viral diseases[1,2] and exosome-mediated cell–cell communication[3,4]) or are promising candidates for novel therapeutic approaches (for example, functionalized vesicles and NPs for targeted drug delivery[5,6]). A full characterization of BNPs requires determination of their size (typically ranging between 20 and 200 nm) as well as the amount of a particular biocompound (for example, expression levels of surface proteins/markers on NPs or DNA/RNA content carried by exosomes). The latter is accessible using appropriate staining procedures, allowing the compound of interest to be quantified based on the emitted fluorescence intensity.

For µm-sized objects such as cells, flow cytometry has proven to be a versatile approach for such measurements[7]. Since its first introduction in 1953 (ref. 8), it has been developed into sophisticated high-end tools for characterization of living cells based on their scattering and fluorescence intensity, allowing now up to 34 different cell parameters to be scrutinized[9]. However, direct application of this technique to smaller objects like BNPs remains challenging as the passage time across the detection region is too short to enable conversion of the weak scattering and fluorescence signals into a quantitative analysis of BNP size, optical density or biomarker content and, importantly, to make correlations between these parameters[10–16].

During the past years, nanoparticle tracking analysis (NTA) has emerged as an attractive alternative to flow cytometry. By light-scattering and/or fluorescence imaging combined with the analysis of the Brownian motion of suspended sub-micron particles, high-precision size distributions of BNPs with radii smaller than 50 nm have been demonstrated[17]. However, since the position of randomly diffusing NPs with respect to the illumination profile varies over time, scattering/fluorescence intensity signals are subject to large variations, causing disturbing fluctuations in plots correlating NP intensity and size[18]. Consequently, it is usually not possible to correlate NP size to optical density and/or specific biomolecular content on the single-NP level.

To reach high accuracy in the determination of the fluorescence intensity emitted by BNPs, Stamou and co-workers[19] developed an assay, in which fluorescently labelled vesicles were immobilized at a glass interface and imaged using confocal microscopy. This setup has the advantage that the BNPs are spatially fixed, allowing their intensity profile to be extracted with high reproducibility. Due to the immobilization, however, Brownian motion cannot be used to extract the BNP size distribution. Hence, two different fluorescent dyes generally have to be used for BNP characterization: one to extract the size distribution[19], and one to detect the biomolecular compound of interest[20]. Furthermore, recent measurements from the same group cast doubts on the validity of this approach, since especially for small vesicles (diameter ∼ 100 nm) large heterogeneities in the dye distribution among the vesicles were observed causing uncertainties in the size determination[21,22].

Accurate BNP characterization would therefore benefit from an approach that sufficiently restricts the BNP movement to allow for an accurate quantification of their emitted scattering or fluorescence intensity, while still permitting a Brownian movement, which can be used for accurate size determination. Herein, we describe such an approach, in which the BNPs of interest are linked to a fluid interface or, more specifically, a lipid bilayer supported at the bottom of a microfluidic channel, while the in-plane BNP movements are recorded using microscopy. Application of hydrodynamic forces on the BNPs induces their drift in flow direction, while the movement perpendicular to the flow remains random, allowing the BNP size to be accurately determined by quantifying the deterministic and random components of the movement. The concept is demonstrated using different linking strategies, all being capable of restricting the movement in two dimensions while keeping the BNPs mobile, and on different BNPs (functionalized gold NPs and small unilamellar vesicles, (SUVs)). Finally, high accuracy in the determination of BNP size and intensity is confirmed by clearly resolving the expected physical dependence between both parameters.

## Results

**Theoretical considerations.** The conventional NTA for determination of NP size distributions using single-particle tracking (SPT) exploits the fact that the bulk-diffusion coefficient $D_b$ of spherical NPs (for example, vesicles, exosomes and so on) within a viscous medium is connected to the hydrodynamic NP radius $R$ by the Stokes–Einstein relation[23],

$$D_b = \frac{k_B \cdot T}{6\pi \cdot \eta \cdot R}, \qquad (1)$$

where $k_B$ is the Boltzmann constant, $T$ the absolute temperature and $\eta$ the dynamic viscosity. The random NP movement is tracked using microscopy and $D_b$ is extracted for each recorded NP trajectory, allowing to create $R$ histograms using equation (1). The measured NP trajectory is usually a two-dimensional (2D) projection of a three-dimensional (3D) movement onto the focal plane of the microscope, as only those NPs can be tracked that are sufficiently close to the focal plane.

As shown in Supplementary Note 1, the main source of random errors in such bulk-based size determination is given by the stochastic noise that is inherent to random walks and has a relative standard deviation $\sigma_R/R$ given by

$$\frac{\sigma_R}{R} = \frac{\langle \Delta R^2 \rangle^{1/2}}{R} = \sqrt{\frac{2}{3} \cdot \frac{N_p}{N - N_p}}, \qquad \text{(for } N_p \ll N) \quad (2)$$

where $N$ denotes the number of frames of the analysed track and $N_p$ the maximum data point separation used in the internal averaging (that is, the product of $N_p$ and the time between two consecutive frames, $\Delta t_0$, gives the maximum lag time involved in the extraction of $D_b$; see Supplementary Note 1 for details). High accuracy in the determination of $R$ therefore requires trajectories covering as many frames $N$ as possible, which is limited, however, by the fact that a NP has to be sufficiently close to the focal plane to be trackable. The average track length can be estimated by (Supplementary Note 1):

$$<N> = \Delta t/\Delta t_0 = z_R^2/2D_b\Delta t_0, \qquad (3)$$

where $z_R$ denotes the depth of focus. Inserting typical values of commercial implementations ($z_R \sim 5\,\mu m$, acquisition rate $1/\Delta t_0 = 30$ f.p.s.)[24] leads to an average trajectory length $<100$ frames for spherical NPs having $R = 50$ nm ($D_b = 4.4\,\mu m^2\,s^{-1}$ in water at 20 °C) and therefore to measurement uncertainties exceeding 9% (equation (2) with $N_p = 1$) for $R$.

The random NP movement through the focal plane obviously limits the accuracy in bulk-based SPT. This can be avoided, however, by restricting the movement of NPs in 2D, for example, by linking them to a fluid interface like a fluid-phase supported lipid bilayer (SLB). This procedure keeps the NPs within the focal plane, thereby increasing the observation time and the accuracy in the measurement of NP properties (for example, its 2D diffusion coefficient, $D$, as well as its fluorescence or scattering intensities; see Supplementary Note 1). Such attempts, however, have not yet been successful for size determinations, since due to

the high friction the linkers experience within the SLB, the $D$ value of the NPs is after linking determined by the used linkers rather than by the NP's hydrodynamic size[25]. Hence, the Stokes–Einstein relation (equation (1)) cannot be applied to extract $R$ from $D$, as done in bulk-based approaches.

As shown here, this apparent shortcoming can be circumvented, if one instead studies linker-constrained 2D NP movement within a microfluidic channel under the influence by a shear flow parallel to the SLB (Fig. 1a). This generates a hydrodynamic shear force $F_s$ acting on the NP in the direction of the flow, causing a NP drift in the flow direction, while perpendicular to the flow the constrained movement remains random (Fig. 1b). In theory, the NP $D$ value and average velocity $v_x$ in the flow direction (denoted as $x$-axis in the following, see Fig. 1b) are connected by the Einstein–Smoluchowski relation[23]:

$$\frac{D}{k_B \cdot T} = \frac{v_x}{F_s}. \tag{4}$$

This relation can be derived from the fluctuation-dissipation-theorem, which states that the forces, causing random fluctuations in the equilibrium state (here, the random forces generated by the diffusing lipids interacting with the linker), also create a dissipation/friction if the system is subject to a non-random force (here, the shear force creating a directed NP movement)[26].

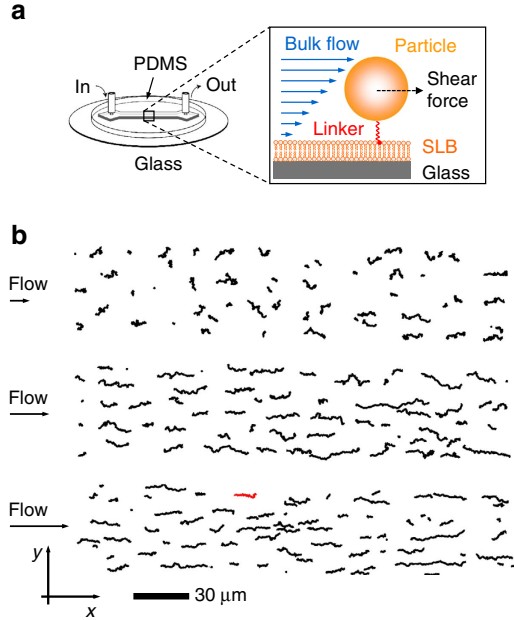

**a**

PDMS
In    Out
Glass

Bulk flow    Particle
                Shear force
Linker        SLB
                Glass

**b**

Flow

Flow

Flow

$y$
$x$    30 μm

**Figure 1 | Size determination of nm-sized objects using 2D flow nanometry.** (**a**) The objects (for example, gold nanoparticles or liposomes) are linked to a fluid interface (for example, a fluid-phase SLB) within a microfluidic channel (using a design as recently described[27]). The linking confines the object's movement into two dimensions, but maintains its ability to move freely. Its movement is monitored from below, for example using scattering, confocal or TIRF imaging. This is demonstrated in **b** for the particular example of streptavidin-functionalized gold nanoparticles (hydrodynamic radius 30 nm) that are linked to biotinylated lipids in the SLB and label-free monitored using surface-enhanced ellipsometric contrast (SEEC) imaging. Application of a flow through the channel creates a shear force, which depends on the flow rate (applied to the channel; 5 μl min$^{-1}$, 10 μl min$^{-1}$ and 15 μl min$^{-1}$ from top to bottom in **b**) and the object's hydrodynamic size. Shown are only the first 100 frames ($=$ steps) of each trajectory corresponding to an observation time of 3 s. The red trajectory is further analysed in Fig. 2.

The derivation of equation (4) implies that the flow-induced motion of the SLB is negligible. This is the case provided that the channel flow rate is below a certain threshold value ($\sim 100$ μl min$^{-1}$ for the channel design shown in Fig. 1)[27], otherwise equation (4) should be complemented by additional parameters[28]. This constraint, however, creates no real limitation, since all flow rates used in this study are far below this threshold. It is also of interest that equation (4) is applicable irrespective of the tiny mechanistic details of NP diffusion, numbers of linkers per NP, and the mechanism of force formation. In our case, as already mentioned, $D$ is determined by the linker–lipid interaction, because for single linkers the diffusion coefficient is typically much smaller ($D \sim 1$ μm$^2$ s$^{-1}$) (refs 25, 29–31) than that for NP in bulk provided $R$ is below 200 nm. In contrast, the value of the drift-inducing force is determined primarily by the NP-solution interaction, while the role of a linker is nearly negligible.

If $v_x$ and $D$ are extractable from single-NP trajectories, application of equation (4) allows us to directly calculate $F_s$ acting on the particular NP. Furthermore, since this force dependends on the NP hydrodynamic radius, it can be used to determine $R$. We will in the following denote this interface-based approach as *2D flow nanometry*. If $z$ is the coordinate perpendicular to the SLB (with the SLB interface at $z = 0$ and $z$ pointing towards the centre of the channel), an analysis shows that $F_s$ scales, in laminar flows, with the product of flow velocity at $z = R$, that is, at the middle of the NP[27,28,32] (provided the linker length is negligible), allowing us to write (Supplementary Note 2)

$$F_s(R) = A \cdot \eta \cdot v_0 \cdot R \cdot (R + \lambda), \tag{5}$$

with $A$ denoting a constant pre-factor (that accounts for the inhomogeneous flow profile around the NP), $v_0$ the flow rate through the channel and the length $\lambda$ specifying the solution behaviour just near the SLB-solution interface (this length is expected to be a few nm; in general, it may include the linker length above the SLB, but in our case this contribution to $\lambda$ is negligible). Note that neither $A$ nor $\lambda$ depend on $R$ or NP type and that both can be determined using calibration measurements (see the subsection 'Calibration using well-defined gold NPs' given below). Consequently, equation (5) can then be employed generally to relate $F_s$ (measured using equation (4)) and $R$, once the used channel design has been calibrated.

**Calibration using well-defined gold NPs.** To test the concept of NP size determination based on 2D flow nanometry, it is most convenient to analyse particles of well-defined size. This was done here using gold NPs, the size distribution of which had been determined by electron microscopy (EM). These NPs were linked via streptavidin to biotin-conjugated lipids in the SLB (Fig. 2a; see the 'Methods' section for details). Due to their high refractive index contrast to the surrounding liquid, SPT of SLB-linked NPs was done label-free using surface-enhanced ellipsometric contrast imaging[33].

Figure 1b shows typical trajectories of gold NPs ($R \approx 30$ nm) measured for flow rates of 5, 10 and 15 μl min$^{-1}$, respectively. The flow rate was observed to increase the rate of the movement in the direction of the flow, which is well reflected in the $x$- and $y$-components of the trajectories (that is, in components parallel and perpendicular to the flow direction). While a predominantly linear increase of the $x$-position was observed (Fig. 2b), indicating a drift in the flow direction, the movement along the $y$-axis indeed appeared to be purely random, as indicated by non-directed fluctuations of the $y$-position displaying no obvious trend (Fig. 2c). The drift was, however, superimposed by fluctuations, causing minor deviations from a perfect linear increase in $x$-position over time (Supplementary Fig. 1). A stringent analysis revealed (Supplementary Note 3) that $v_x$

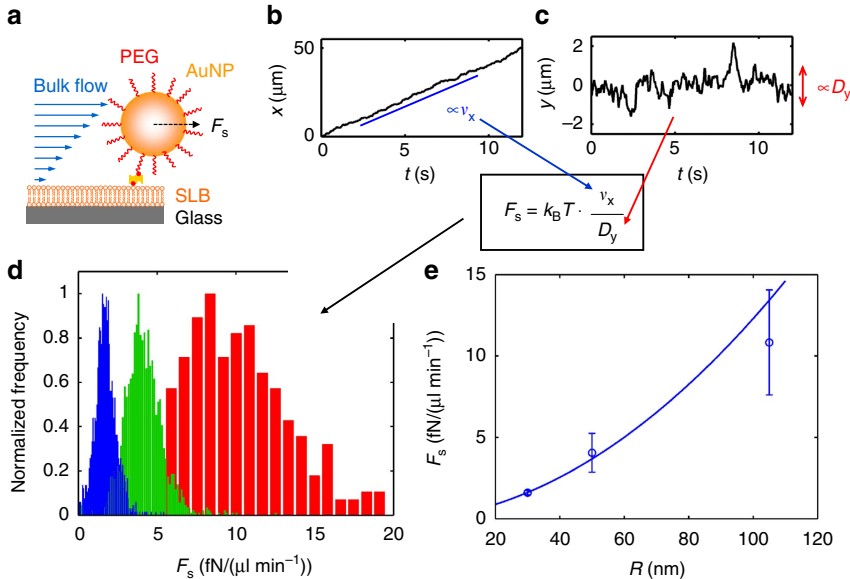

**Figure 2 | Calibration using well-defined gold nanoparticles.** Streptavidin-functionalized gold nanoparticles (**a**; with a PEG shell thickness of 5 nm) are linked to biotinylated lipids (linker length of ∼5 nm) in the SLB and label-free monitored using surface-enhanced ellipsometric contrast (SEEC) imaging. (**b,c**) give a decomposition of a representative trajectory (30 nm gold nanoparticle; 15 µl min$^{-1}$ flow rate) into its component in flow direction (x-axis) and perpendicular to the flow (y-axis). Due to the flow, the x-component is dominated by a directed movement (indicated by its linear increase with time), allowing to extract the induced velocity $v_x$, while the y-component remains (due to absence of shear force in this direction) fully random and allows to extract the linker diffusion coefficient $D_y$ (see Supplementary Note 3 for details). Combining both information yield the hydrodynamic shear force $F_s$ acting on the particular nanoparticle. Histograms (**d**) of $F_s$ (after normalization to the flow rate) exhibit a peak at 1.60 fN min µl$^{-1}$ for 30 nm (blue), at 4.05 fN min µl$^{-1}$ for 50 nm and at 10.83 fN min µl$^{-1}$ for 105 nm gold nanoparticles (hydrodynamic radius). These calibration measurements allowed to fit equation (5) (**e**), which is required to convert distributions of the hydrodynamic force into size distributions. The solid line in **e** gives the result of a weighted least squares fit that also takes the standard deviation of the $F_s$ distribution (error bars) into account and yields $\lambda = 24.4$ nm and $A\eta = 1$ fN min µl$^{-1}$.

can be extracted from the linear increase of the x-position, while the diffusion coefficients in the x- and y-directions, $D_x$ and $D_y$, can be determined from the fluctuations of the x- and y-components, respectively. Since the SLB is a 2D isotropic medium, $D_x$ and $D_y$ are expected to be equal, which is (within experimental error) observed for the gold NPs (Supplementary Fig. 2), demonstrating that the data-extraction procedure successfully decouples the directed and random NP movements and allows calculating the 2D diffusion coefficient $D$ (as an arithmetic average of $D_x$ and $D_y$) and $F_s$ using equation (4).

Furthermore, equations (4 and 5) suggest that $v_x$ scales linearly with $v_0$ and $D$, which is consistent with our observations (Supplementary Fig. 3). Hence, after normalizing $v_x$ by the applied flow rate $v_0$, all data points collapse onto a single master curve (red line in Supplementary Fig. 3d), allowing to quantitatively relate experiments performed at different $v_0$ by regarding rather the normalized velocity $v_x/v_0$ instead of $v_x$ itself. Note that the noise in Supplementary Fig. 3 decreased with increasing $v_0$, which is attributed to the fact that higher flow rates induce larger NP displacements between consecutive frames. This in turn increases the signal-to-noise ratio in the measurement of $v_x$. Hence, the random error in the determination of $F_s$ can be reduced by increasing $v_0$ (as suggested in Supplementary Note 1), an optimization strategy that is not supplied by bulk-based approaches.

As already noted, substituting $v_x$ and $D$ into equation (4) makes it possible to directly extract $F_s$ acting on each tracked NP. Fig. 2d shows histograms of the normalized hydrodynamic force, $F_s/v_0$, measured for NPs having average hydrodynamic radii of 30, 50 and 105 nm, exhibiting peaks in the $F_s/v_0$-distributions at 1.60, 4.05 and 10.83 fN min µl$^{-1}$, respectively. These measurements allowed determining the calibration

parameters $A$ and $\lambda$ in equation (5) (Fig. 2e, solid line), and therefore to calibrate the microfluidic channel for the determination of full size distributions yielding $\lambda = 24.4$ nm and $A\eta = 1$ fN min µl$^{-1}$ (see also Supplementary Note 4 for detailed description and discussion of the calibration procedure).

This is further demonstrated in Fig. 3 and Supplementary Fig. 4 comparing size distributions obtained from 2D flow nanometry and EM. Both methods yielded essentially the same distributions if a shift of 5 nm is taken into account, which is attributed to the PEG corona formed on the NP surface that is not resolvable in the EM images[34]. Repetition of such measurements indicated high reproducibility (Supplementary Fig. 5a). In addition, linking the gold NPs specifically to transmembrane proteins (using an antibody-functionalized PEG corona as recently described by Johansson Fast[35]) instead of using biotin–streptavidin-links did not affect the extracted distributions (Supplementary Fig. 5b), indicating also minor influence of the particular linking strategy.

**Correlation of size and fluorescence intensity of SUVs.** For further validation of the proposed 2D flow nanometry approach, it was applied to analyse sub-100 nm vesicles (Fig. 4). The vesicles (fluorescently labelled by incorporation of lissamine rhodamine-conjugated lipids) were linked to the SLB using cholesterol-conjugated DNA-tethers (Supplementary Fig. 6 and Supplementary Note 5) and tracked using total internal reflection fluorescence microscopy (TIRFM), as previously described[25,29]. A good agreement between the 2D flow nanometry size distributions and those obtained with NTA (Fig. 4b) and dynamic light scattering (DLS, Fig. 4c) was observed (see also Supplementary Figs 7 and 8). Interestingly, while the NTA size

distribution showed no vesicles with hydrodynamic radii below 25 nm (Fig. 4b), such vesicles were resolved using 2D flow nanometry and DLS (Fig. 4c). This is one order of magnitude better than reported for commercial flow cytometers[10–16].

In contrast to DLS, which does not offer single-vesicle resolution, 2D flow nanometry and NTA allow information about vesicle size and fluorescence intensity to be extracted (Fig. 5). However, the intensity traces extracted with these methods differed considerably, which is attributed to the fact that in 2D flow nanometry the tracked vesicles remain within the focal plane during their passage through the field of view, leading to stable intensity traces (Fig. 5a), while in NTA the vesicles were free to move in 3D and therefore continuously enter and exit the focal plane, leading as expected (especially for small vesicles) to

strong fluctuations of $I$ (Fig. 5b). Linking the vesicles to a fluid interface thus led to much better defined $I$ values, which is well reflected in a plot of $R$ versus $I$ for both vesicle batches (Fig. 5c,d). In particular, the NTA data points (grey dots) showed strong fluctuations, making it impossible to resolve the expected scaling ($I \propto R^2$, solid line) in this parameter plot. For 2D flow nanometry (black dots), much lower fluctuations were observed and the expected scaling ($I \propto R^2$, solid line)[36] is clearly resolved in the range down to 15 nm, illustrating that maintaining vesicles in the focal plane during the whole measurement enables accurate determination of $I$.

## Discussion

A new approach was introduced for the size determination of NPs, which are linked to a fluid interface within a microfluidic channel. A shear force (generated by a channel flow) induced a directed NP movement, while the movement perpendicular to

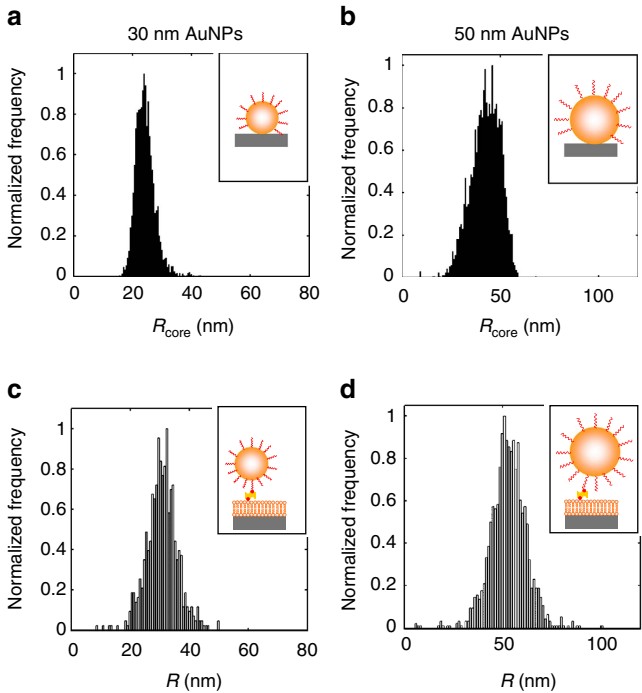

**Figure 3 | 2D flow nanometry applied to gold NPs.** Comparison of size distributions of gold NP batches determined using EM (**a,b**) and 2D flow nanometry (**c,d**). The distributions are essentially identical and are only shifted by 5 nm, which is attributed to a 5 nm PEG corona formed on the surface of the gold nanoparticles that is not resolved in EM[34]. Note that the widths of the distributions are indicative for the polydispersity of the gold NP samples, but not the measurement accuracy of the approaches, which is on the nm-scale.

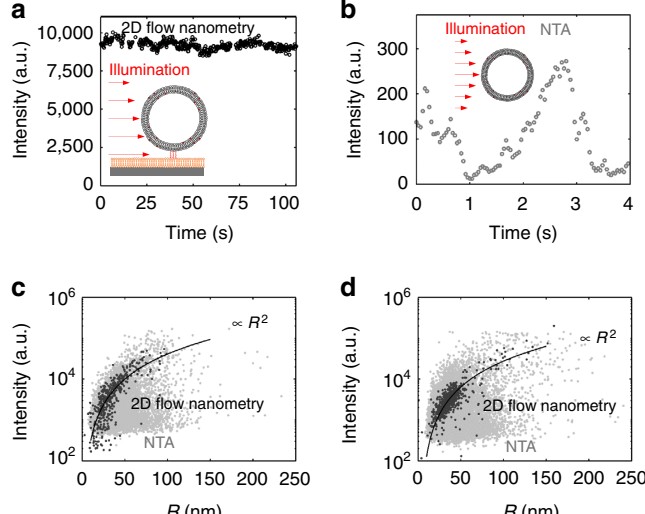

**Figure 5 | Comparison of vesicle intensity extraction done by 2D flow nanometry and NTA. a** and **b** show representative intensity traces for a single, fluorescently labelled vesicles, while **c** and **d** compare the intensity-versus-size parameter plots obtained by NTA (grey dots) and 2D flow nanometry (black dots) for two different batches of SUVs (created by sonication (**c**) and extrusion (**d**) as described in the 'Methods' section). Both batches show peak hydrodynamic radii around 38 nm, but differ in their polydispersity. Due to much lower intensity fluctuations observed in 2D flow nanometry (**a**) with respect to NTA (**b**), the expected scaling law is well visible in the parameter plots (solid lines in **c,d**), while it is hard to resolve for NTA data.

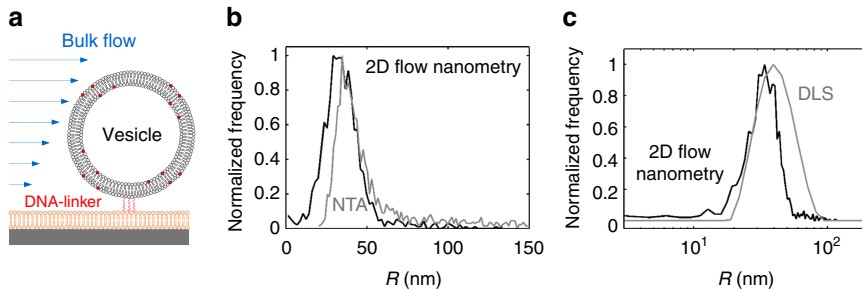

**Figure 4 | 2D flow nanometry applied to SUVs.** The SUVs (created by extrusion as described in the 'Methods' section) were linked by cholesterol-equipped 13 nm DNA-tethers to the SLB (**a**). Comparison of vesicle size distributions obtained by NTA (**b**, grey curve), DLS (**c**, grey curve; intensity distribution) and using 2D flow nanometry (**b,c**, black curve).

this shear force remained random. This allowed to directly extract the shear force acting on the NP and therefore its hydrodynamic size. The approach is versatile (see Supplementary Note 6 for present limitations), which was demonstrated by successful application on inorganic (gold NPs) and biological NPs (fluorescent SUVs), using label-free detection based on surface-enhanced ellipsometric contrast imaging and TIRFM, respectively. In contrast to the well-established bulk NTA approach, linking confines the NP movement within the focal plane in 2D flow nanometry, enabling also accurate extraction of the emitted NP intensity.

The NP passage time through the field of view can be adjusted by the channel flow rate, thereby allowing optimizing the measurement accuracy. Other size-determination approaches lack such an optimization strategy. The flow rate should be chosen in such a way that the distance travelled by the directed movement, $v_x \cdot \Delta t_0$, is larger than the square root of the mean squared displacement, that is, $v_x > \sqrt{4 \cdot D / \Delta t_0}$. As $D$ is given by the diffusion properties of the linker (and not of the NP itself), which is typically on the order of $1 \, \mu m^2 \, s^{-1}$, this suggests that $v_x$ exceeds $5 \, \mu m \, s^{-1}$ for typical acquisition rates of $1/\Delta t_0 \approx 10 \, \text{f.p.s.}$ Using equation (4) this in turn indicates that the shear force acting on the NPs should be larger than 25 fN, a value which is independent of the NP size as it follows from the diffusive properties of the linkers and acquisition rate of the microscope. Referring back to Fig. 2e, this shows that (in this study) a flow rate of $15 \, \mu l \, min^{-1}$ is sufficient for NPs with $R$ as small as 20 nm, which is far below the threshold to induce directed lipid movement in the SLB[27].

Further, the field of view of the microscope used in this study ($x$-extension $\sim 200 \, \mu m$) was large enough to track many of the gold NPs for at least 150 frames and most of the vesicles for at least 200 frames. The difference in these numbers is caused by the slightly lower $D$ value of the vesicles due to the different linking strategy. This leads (Supplementary Note 1) to expected relative random errors $\sigma_R / R$ of 5% for the gold NPs, corresponding to a size accuracy of $\pm 2 \, nm$ for the batches in Fig. 3c,d. These values seem to be reasonable since the size distributions extracted with 2D flow nanometry and EM are very similar and since the latter is known to offer nm-resolution for metal NPs.

For vesicles, the expected relative random errors are even lower with $\sigma_R / R = 4\%$, translating into $\pm 1.5 \, nm$ for the batches shown in Fig. 5. Such high accuracies are indeed necessary to resolve the $R$-$I$ relationship (Fig. 5c,d), since increased fluctuations would otherwise smear out the data points into a point cloud. We therefore tried to suppress such fluctuations in NTA-derived $R$-$I$ plots by increasing the minimum number of frames required for trajectories to be included in the data analysis. This indeed reduced the fluctuations, but still did not permit to unambigeously resolve the expected $R$-$I$ relationship.

Motivated by Larsen et al.[22] we calculated the ratio of measured and expected intensity (Fig. 5c,d; black dots versus solid line) to assess heterogeneities in dye-labelled lipid distributions across individual vesicles (Supplementary Fig. 9). The s.e. of this ratio is 0.41 and 0.30 (for Fig. 5c,d, respectively), which is lower than expected from ref. 22, since the standard deviation increases with decreasing $R$ and since Larsen et al.[22] report similar values, although for much larger vesicles (radii ranging between 50 and 400 nm).

Note that the length $\lambda$ in equation (5), (connecting $R$ and $F_s$) was found to be important, since attempts to fit equation (5) failed for $\lambda = 0$. In particular, the size distribution of the 30 nm gold NP batch was systematically overestimated, while the ones of the larger gold NP batches and all vesicle batches became systematically underestimated. The extracted $\lambda$ value of 24.4 nm is somewhat larger than intuitively expected, but on the same order of magnitude as in recent reports[37–39]. Including also the SUV

data sets into the calibration procedure, which was initally restricted to the gold NP data sets, yielded essentially the same fitting parameters (Supplementary Fig. 10), indicating self-consistency of the approach. Further, flow-induced vesicle deformation was not observed (Supplementary Fig. 11 and Supplementary Note 7).

In summary, our experiments have shown that the concept of 2D flow nanometry makes it possible to combine 2 different approaches, which seemed to be incompatible in the past, that is, sufficiently restricting the NP movement to allow for accurate measurements of NP emission, while still permitting them to obey a Brownian motion, which is required for accurate measurements of their size. Although so far demonstrated on two generic examples, application to other BNPs like viruses, micelles and exosomes appears to be feasible and marks the next steps.

## Methods

**Materials.** POPC (1-palmitoyl-2-oleoyl-sn-glycero-3-phosphocholine), DSPE-PEG(2k)biotin (1,2-distearoyl-sn-glycero-3-phosphoethanolamine-N-[biotinyl (poly(ethylene glycol))-2000] and rhodamine-DOPE (1,2-dioleoylsn-glycero-3-phosphoethanolamine-N-(lissamine rhodamine B sulfonyl)) were obtained from Avanti Polar Lipids Inc. (Alabaster, AL, USA). Cholesterol-terminated DNA strands were obtained from Eurogentec S.A. (Seraing, Belgium) with the following sequences: 5'-TGGACATCAGAAATAAGGCACGACGGACCC-chol-3' ($\alpha$); 5'-chol-CCCTCCGTCGTGCCT-3' ($\alpha'$); 5'-TATTTCTGATGTCCAAGCCACG AGTTCCCC-chol-3' ($\beta'$); 5'-chol-CCCGAACTCGTGGCT-3' ($\beta$). Tris(hydroxymethyl)-aminomethane hydrochloride (TRIS-HCl), sodium chloride and calcium chloride were obtained from Sigma Aldrich (Steinheim, Germany). $\alpha$-Hydroxy-$\omega$-mercapto-PEG, $\alpha$-carboxy-$\omega$-mercapto-PEG and $\alpha$-biotinyl-$\omega$-mercapto-PEG (all having a molecular weight of 5 kDa) were purchased from RAPP Polymere (Tübingen, Germany). If not otherwise stated, all solutions were prepared or diluted using a TRIS-HCl buffer consisting of 100 mM Tris-HCl, 50 mM NaCl, 5 mM $CaCl_2$ that was adjusted to pH = 7.4 using HCl.

**Vesicle preparation.** SUVs were prepared by the extrusion method as described earlier[40], or, alternatively, by sonication. For extrusion, lipid films were formed in round-bottom flasks under flowing nitrogen and dried in vacuum, hydrated by adding 1 ml of the Tris-HCl buffer, followed extruding the mixture through polycarbonate membranes (Avanti Polar Lipids Inc.). Alternatively, to produce SUVs with different size distribution, these were formed by sonicating the lipid–buffer mixture (contained in a test-tube immersed in an ice-bath) using a tip sonicator for five times 5 min.

Vesicles for SLB formation consisted of 99.8 mol% POPC and 0.2 mol% DSPE-PEG(2k)biotin, and were extruded (pore size 100 nm), while for SPT two vesicle batches were created having a composition of 97 mol% POPC + 3 mol% rhodamine-DOPE (sonicated; shown in Fig. 5c) or of 98 mol% POPC + 2 mol% rhodamine-DOPE (extruded with pore size 50 nm; shown in Fig. 5d).

**Gold nanoparticles.** Gold NPs with a hydrodynamic radius of 30 nm were synthesized by seed-mediated growth according to a modified version of the protocol presented by Park et al.[41] using ascorbic acid as a reducing agent. The larger gold NPs were synthesized using the protocol presented by Perrault et al.[42] with hydroquinone as the reducing agent. Gold NPs were surface functionalized by chemisorption of thiolated poly(ethylene) glycol (PEG) ligands from a mixture of $\alpha$-hydroxy-$\omega$-mercapto-PEG, $\alpha$-carboxy-$\omega$-mercapto-PEG and $\alpha$-biotinyl-$\omega$-mercapto-PEG in water solution. With the aim to modify each gold NP with a single biotinylated ligand, the relative content of $\alpha$-biotinyl-$\omega$-mercapto-PEG in the mixture was adjusted for the different particle sizes in relation to their different surface areas. This was done assuming an approximate grafting density of $1 \, nm^{-2}$ (independent of NP size) for the different thiolated ligands, which means that for NPs with radius $\sim 25 \, nm$ and corresponding surface area $\sim 8,000 \, nm^2$, the content of $\alpha$-biotinyl-$\omega$-mercapto-PEG was 1/8,000 relative the total content of thiolated PEG in the mixture. After surface modification, NPs were purified from excess ligand by filtration using centrifuge filter columns with 300 kDa-cutoff (PALL, USA). The gold NPs were further conjugated with streptavidin by adding gold NPs to a solution containing streptavidin in excess, followed by filtration as described above.

NP size distributions were determined by transmission EM. Electron micrographs were recorded on a FEI Tecnai G2 microscope operated at 160 kV acceleration voltage. The gold NPs were applied on formvar and carbon-coated cupper grids (FCF300-Cu-TB, Electron Microscopy Sciences, USA). These samples were hydrophilized by ultraviolet/ozone treatment for 5 min using an UV/Ozone ProCleaner from Bioforce Nanoscience and then further treated with poly-L-lysine, which was applied by positioning the grid upside-down on a small droplet of poly-L-lysine solution ($20 \, \mu g \, ml^{-1}$ in MilliQ water). Such-treated grids were coated with gold NPs by first positioning grids upside-down on top of droplets with

gold NP suspension (concentrated by centrifugation) for 15 min whereupon grids were blotted on a filter paper.

**TIRF-Microsopy.** TIRFM was performed on an inverted Eclipse Ti microscope (Nikon, Japan) that was equipped with a high-pressure mercury lamp, an Apo TIRF $60\times$ oil objective (NA 1.49), and an Andor Neo CCD camera (Andor Technology, Belfast, Northern Ireland). A rhodamine filter set (TRITC, Semrock, Rochester, NY, USA) was used, while focus drift was effectively reduced using the microscope's Perfect Focus System.

**SLB formation and vesicle tethering.** All TIRF experiments were performed on glass microscope coverslips as surfaces, which were supplemented with a home-made polydimethylsiloxane microfluidic channel (using the design as recently described[27]). SLBs were formed by injecting POPC vesicles (0.1 mg ml$^{-1}$, flow rate 20 µl min$^{-1}$ for 20 min), followed by rinsing with the Tris-HCl buffer (flow rate 20 µl min$^{-1}$ for 20 min). Vesicles were linked to a SLB using cholesterol-modified DNA strands as described earlier[25,29]. In brief, SLBs and vesicles were incubated separately with two different types of DNA strands, which carry a double-cholesterol group at one end that self-inserts the strands into the lipid bilayers. Both types of DNA strands share a conjugated single-stranded part at the other end, which allows linking vesicles to the SLB via hybridization (see Supplementary Note 5 for details).

**Data analysis.** All data analysis was done using home-made scripts written in MatLab (MathWorks, Natick, MA, USA). SPT was implemented using local nearest-neighbour linking[43]. Diffusion coefficients were calculated using the internal averaging procedure[44] with a maximum data point separation $N_p = 2$ as described in Supplementary Notes 1 and 3 and corrected for motion blur[45].

**Data availability.** The data that support the findings of this study are available from the corresponding authors on request.

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

## Acknowledgements

We thank the Knut and Alice Wallenberg Foundation (2012.0055), the Swedish Research Council (2014–5557) and the Swedish Foundation for Strategic Research (RMA11-0104) for funding.

## Author contributions

B.J.F. did the microfluidic experiments with support from S.B. (vesicle preparation and linking) and A.L. (synthesis, functionalization, EM characterization of gold NPs and vesicle preparation); S.B. developed the theoretical analysis with help from V.P.Z.; S.B. analysed the experimental data; S.B. and F.H. conceived the project and wrote the paper together with V.P.Z., B.J.F. and A.L.

## Additional information

**Competing financial interests:** S.B., B.J.F., A.L. and F.H. submitted a patent application containing information presented in this study. V.P.Z. declares no competing financial interests.

