## [Peer Review File · Nature Communications]

Reviewers' comments:

Reviewer #1 (Remarks to the Author):

In this interesting article the authors propose and demonstrate a new approach to characterising "biological nanoparticles", for example vesicles such as exosomes, and other structures in the tens to hundreds of nanometres size range.

As they set out very clearly in their introduction, the size (from light scattering) and composition (from fluorescence intensity) can be measured for larger objects - such as cells - by flow cytometry. However this approach lacks the sensitivity to accurately measure smaller particles.

Therefore, the radius of nanoparticles in suspension is typically measured by tracking their Brownian motion (as in NTA or Nanoparticle Tracking Analysis), where freely diffusing particles are imaged. However this approach has a significant limitation. Because the particles are diffusing in three dimensions, but are only imaged in two, they may diffuse out of the imaging plane, which limits the length of trajectories that may be observed, and therefore the precision with which their radii may be measured. Furthermore, this diffusion in the third dimension also affects measures of fluorescence intensity.

The deceptively simple solution which the authors propose constrains the particles to diffusion in 2D, and applies a shear force. The constraint to the imaging plane allows longer trajectories to be observed and therefore more accurate particle sizes - and also more accurate fluorescence intensities.

The constrained diffusion is achieved by linking the particles to a supported lipid bilayer. They show how lateral diffusion is dominated by the properties of the membrane linker, whereas applying shear force through flow in a microfluidic channel allows measurement of the particle size.

Broadly I think this is an interesting paper, presenting a method which could gain wide acceptance by providing a solution to a common measurement challenge. The text, for the most part, is written with exemplary clarity and the figures are a model of intelligibility. I believe this to be an original solution to this measurement challenge, and as such it will be of interest to a wide range of readers.

General comments:

An aspect that the authors touch on only very briefly are the effects of the linkers. A criticism that is made very often of fluorescence approaches is that they may change the properties or behaviour of the entity measured. In this case one might imagine that the linkers could cause aggregation of some of the particles, particularly the lipid vesicles. Is this ever observed, and if not, why not?

How can the authors exclude the possibility that there is more than one linker between a particle and the bilayer? (As in fact is illustrated in Figure 5a) Would this be apparent from the data? Does the measurement of D_y for each particle allow this effect to be removed?

Specific comments:

Under "Results" "Theoretical considerations" the selection of lag times is not explained. This section could be clarified as N_p is explained by reference to an equation that includes... N_p !

As I understand it the "Rayleigh Length" relates to Gaussian beams rather than imaging - I think depth of focus would be the more usual term. However 5 μm sounds reasonable for a high NA

objective.

I found $D_b = 4.9 \mu\text{m}^2/\text{s}$, rather than 4.4, for the scenario proposed, using $k_B = 1.38 \times 10^{-23} \text{ J/K}$ and $\eta = 0.8902 \text{ mPa}\cdot\text{s}$ at 25 {degree sign}C. Have I missed something?

"For typical linkers the diffusion coefficient ($< 1 \mu\text{m}^2/\text{s}$) is much smaller than that for NP in bulk provided R is below 200 nm." What will be the practical effect as R approaches 200 nm? Is it possible to account/compensate for these effects? This is a significant point as many naturally occurring populations of particles - for example - exosomes - will have a very broad size distribution and include some larger particles. I think it's important to be clear about the limitations of the proposed approach.

Figure 2: Please define LMS.

Figure 3: letters referring to the figure panels appear to be incorrect. Surely EM is a, b and flow nanometry is c, d?

Supporting information:

Section 1 duplicates a lot of the discussion in the main text and could be pruned accordingly.

Figure 1e: solid line for average value is missing.

Supporting Movie 1: What causes background of static (non-moving) particles? How are these particles removed from the analysis? While it is always dangerous to interpret such stochastic phenomena by eye, some of the particles appear to "stick" (i.e. stop diffusing) for short periods. Is this impression borne out by the tracking data, and what effect does it have on the results?

Reviewer #2 (Remarks to the Author):

see attached file

Reviewer #3 (Remarks to the Author):

The authors introduced a 2D flow cytometry device for separating nanoparticles. This reviewer cannot recommend the publication of this manuscript in Nature Communications due to lack of novelty and several technical issues.

1. Authors state that flow cytometry is incapable of fast detection of weak scattering of fluorescence signals to quantify the biological nanoparticles (BNPs). But this statement is not well supported by references. Actually, many different flow cytometry methods have been reported over the years for BNP measurements.
2. Figure 2e. Only three sizes of gold nanoparticles were used in this calibration. Due to large error bars, this calibration is not reliable. Gold nanoparticles of different sizes (e.g., 60 and 80 nm) should also be measured.
3. Page 13, line 292 and 297. The authors state that the size accuracy of {plus minus} 2 nm for the gold NP and {plus minus} 1.5 nm for vesicles were achieved using 2D flow nanometry. However, the histograms of the size distributions in Figure 3 are clearly broader than what was stated.
4. In Figure 4b and 4c, the band width of 2D flow nanometry is actually larger than that of NTA and DLS.
5. SI Movie 1 "95337_0_video_1733796_p6pr1x.avi". How could the authors determine whether a

moving particle is bound to the surface receptors in the supported lipid bilayer? Non-specific interactions can be another reason for the nanoparticles to stay on the lipid bilayer. Furthermore, if the particles bound to supported lipid bilayer move with the applied shear flow force, how to explain those particles that are not moving? Many of those stationary particles have strong signal, which may suggest that these particles are relatively large in size.

Other comments:

Page 4, line 64 to 67: The BNPs linked to SLB still performing Brownian movement perpendicular to the flow direction. It should be interesting to quantify the effects of hydrodynamic forces on the magnitude of the Brownian motion.

Page 11, line 235 and 236: How will the morphology of the vesicles change in response to the hydrodynamic shear flow forces?

In Page 8, line 170 and Page 9, line 173, the authors stated that length λ is negligible. However, in Page 14, line 311 and 312, they make a different point: "Note that the length λ in Eq. 5, (connecting R and F_s) was found to be important, since attempts to fit Eq. 5 failed for $\lambda = 0$." Legend for Figure 3 (Page 22) is incorrect. Figure 3a and 3b are electron microscopy measurements, while Figure 3c and 3d are 2D flow nanometry measurements. The references have inconsistent formats.

We thank the reviewers for their valuable comments and suggestions regarding the manuscript, which we have carefully accommodated (see below). Major changes of the main text and the supporting information have been marked in red.

Reviewer #1

In this interesting article the authors propose and demonstrate a new approach to characterising "biological nanoparticles", for example vesicles such as exosomes, and other structures in the tens to hundreds of nanometres size range.

As they set out very clearly in their introduction, the size (from light scattering) and composition (from fluorescence intensity) can be measured for larger objects - such as cells - by flow cytometry. However this approach lacks the sensitivity to accurately measure smaller particles.

Therefore, the radius of nanoparticles in suspension is typically measured by tracking their Brownian motion (as in NTA or Nanoparticle Tracking Analysis), where freely diffusing particles are imaged. However this approach has a significant limitation. Because the particles are diffusing in three dimensions, but are only imaged in two, they may diffuse out of the imaging plane, which limits the length of trajectories that may be observed, and therefore the precision with which their radii may be measured. Furthermore, this diffusion in the third dimension also affects measures of fluorescence intensity.

The deceptively simple solution which the authors propose constrains the particles to diffusion in 2D, and applies a shear force. The constraint to the imaging plane allows longer trajectories to be observed and therefore more accurate particle sizes - and also more accurate fluorescence intensities.

The constrained diffusion is achieved by linking the particles to a supported lipid bilayer. They show how lateral diffusion is dominated by the properties of the membrane linker, whereas applying shear force through flow in a microfluidic channel allows measurement of the particle size.

Broadly I think this is an interesting paper, presenting a method which could gain wide acceptance by providing a solution to a common measurement challenge. The text, for the most part, is written with exemplary clarity and the figures are a model of intelligibility. I believe this to be an original solution to this measurement challenge, and as such it will be of interest to a wide range of readers.

We sincerely thank the Reviewer for his/her positive opinion about our work and constructive comments and suggestions, which are addressed below.

General comments:

I.1. An aspect that the authors touch on only very briefly are the effects of the linkers. A criticism that is made very often of fluorescence approaches is that they may change the properties or behaviour of the entity measured. In this case one might imagine that the linkers could cause aggregation of some of the particles, particularly the lipid vesicles. Is this ever observed, and if not, why not?

ANSWER: In our case, linker-induced aggregation is not observed, which is a consequence of the self-assembly strategy and the properties of the DNA linkers. The linking strategy involves incubating the supported lipid bilayer (SLB) and vesicles with 2 different types of double cholesterol-terminated DNA strands, which share a complementary region at their cholesterol-free end (Supporting Figure 5 in the revised version). It has been shown that the double cholesterol-terminated DNA strands have high inserting efficiency and are firmly bound to the bilayers on the timescale of the experiment (I. Pfeiffer & F. Höök. *Analytical chemistry* 2006, 78, 7493). Hence, during incubation the majority of linkers will self-insert via the double cholesterol group into the bilayers and remain there. Injecting DNA-modified vesicles to the DNA-modified SLB (formed within the PDMS channel) allows sequence-specific hybridization of vesicle and SLB DNA, thereby linking the vesicles to the SLB. Linker-induced

aggregation requires that SLB DNA-strands have been inserted into the vesicles, which can be ruled out due to excessive rinsing of the PDMS channel with buffer prior to vesicle incubation and due to stable insertion of DNA-strands into the bilayers, which is practically irreversible on the timescale of the experiment, making DNA transfer between the SLB and the vesicles negligible.

In principle, the linkers themselves may tend to aggregate in the SLB and it might result in the vesicle aggregation. This mechanism implies aggregation of linkers belonging to different vesicles. Under our conditions, each vesicle has only a few linkers (see point I.2. below), and the aggregation of linkers belonging to different vesicles is energetically unfavourable because it may occur only if vesicles are appreciably deformed (this costs energy).

There have been attempts in the past to study DNA-mediated aggregation of vesicles linked to a SLB (Y. H. M. Chan, B. van Lengerich & S. G. Boxer. *Proceedings of the National Academy of Sciences* 2009, 106, 979). In this case, the vesicles have been modified with 2 different types of linkers, one type to enable linking of the vesicles to the SLB and a second one to mediate sequence-specific hybridization of adjacent vesicles. Even in presence of these vesicle-vesicle linkers only low aggregation rates have been observed, which is attributed to the low collision rate above the SLB, caused by the low vesicle surface coverage required for single particle tracking. This further indicates that aggregation can be completely neglected in absence of such vesicle-vesicle linkers.

ACTION: We thank the reviewer for making us aware that the linking procedure is insufficiently described. We therefore added a new section to the supporting information (Supporting Section 6) containing details and discussions regarding vesicle linking. In that section, we explain that the linker-mediated aggregation of vesicle is unlikely.

I.2. How can the authors exclude the possibility that there is more than one linker between a particle and the bilayer? (As in fact is illustrated in Figure 5a) Would this be apparent from the data? Does the measurement of D_y for each particle allow this effect to be removed?

ANSWER: The reviewer is right that nanoparticles (NPs) can be linked to the SLB via multiple linkers and that the measurement of D_y allows this effect to be removed. This is implemented in the data analysis by Equation 4 in the main text, taking the ratio of the diffusion coefficient and the velocity v_x of flow-induced directed movement to directly obtain the flow-induced hydrodynamic force F_s without the requirement of knowing details of the linking strategy (as mentioned on page 8 of the main manuscript). Note that we use the 2D diffusion coefficient D in Equation 4, which is in isotropic systems equal to the 1D diffusion coefficients D_x and D_y , but can be measured with higher accuracy (see Supporting Section 3). This holds as long as the NP mobility is dominated by the mobility of the linker, i.e., as long as the friction coefficient of the linkers within the SLB is much larger than the friction coefficient of the NP within the solution. In this case, v_x and D are both determined by the linkers and are quantitatively connected via the fluctuation-dissipation theorem given by Equation 4. Further, D becomes independent of the actual NP size, since the solvent friction can be neglected in comparison to the linker friction within the SLB. NPs that are connected by a larger number of

linkers will have a lower D , but (according to Equation 4) also a lower v_x , allowing to remove effects caused by differences in the number of linkers by taking the ratio of v_x and D . We see this feature as one of the main advantages of our approach, since it is not necessary to know the actual dependence of the number of linkers on D , which may vary for different linking strategies. As long as the NPs are linked while remaining a 2D mobility, F_s can be immediately extracted from the data, making our approach very versatile.

ACTION: We included additional information in the discussions on the applicability of Equation 4 on page 8 (marked in red).

Specific comments:

I.3. (i) Under "Results" "Theoretical considerations" the selection of lag times is not explained. (ii) This section could be clarified as N_p is explained by reference to an equation that includes... N_p !

ANSWER:

(i) We thank the reviewer for this valuable suggestion. The selection of lag times varies between the different practitioners of single particle tracking and is in some cases even undisclosed (e.g., for the analysis software of most commercial NTAs). We therefore did not further specify the values used for N_p on page 6, as the considerations are general and not bound to a particular implementation. But we fully agree that the selection of N_p should be explained for our settings, which is now included in the Materials and Methods section and Supporting Section 1.

(ii) We also agree that it is not optimal to introduce the parameter N_p by referring to an equation that includes N_p , which was a consequence of the fact that it is not easy to define N_p in a single sentence without giving details of the so-called internal averaging method. Giving these details would improve the clarity of this section, but also add a lot of technical details, which are actually not necessary for an understanding of 2D flow nanometry. We therefore decided to rephrase the corresponding section and refer to Supporting Section 1 for the technical details of the internal averaging method, with the hope that the clarity of the presentation is improved by this step.

ACTION: We rephrased accordingly and added details of the internal averaging procedure to Supporting Section 1 (marked in red).

I.4. As I understand it the "Rayleigh Length" relates to Gaussian beams rather than imaging - I think depth of focus would be the more usual term. However 5 μm sounds reasonable for a high NA objective.

ACTION: We thank the reviewer for this suggestion and rephrased accordingly on page 6 in the main text and page 3 in the Supporting Information.

I.5. I found $D_b = 4.9 \mu\text{m}^2/\text{s}$, rather than 4.4, for the scenario proposed, using $k_B = 1.38 \times 10^{-23} \text{ J/K}$ and $\eta = 0.8902 \text{ mPa}\cdot\text{s}$ at 25 {degree sign}C. Have I missed something?

ANSWER: We agree and thank the reviewer for spotting this typo. We calculated D_b at $T = 20^\circ\text{C}$, which is the temperature the experiments have been performed at, and not at $T = 25^\circ\text{C}$ as indicated in the text.

ACTION: We corrected the temperature on page 6.

I.6. "For typical linkers the diffusion coefficient ($< 1 \mu\text{m}^2/\text{s}$) is much smaller than that for NP in bulk provided R is below 200 nm." What will be the practical effect as R approaches 200 nm? Is it possible to account/compensate for these effects? This is a significant point as many naturally occurring populations of particles - for example - exosomes - will have a very broad size distribution and include some larger particles. I think it's important to be clear about the limitations of the proposed approach.

ANSWER: The approach in its current version requires that the diffusion is limited by the linker, i.e., the friction coefficient of the linkers within the SLB exceeds the friction coefficient of the NP within the solution. As the diffusion coefficient of a single linker is usually on the order of $1 \mu\text{m}^2/\text{s}$, this constrain is expected to be broken if R approaches 200 nm. In this case, the velocity of flow-induced directed movement will become comparable or equal to the fluid velocity at the NP's midplane, i.e., the NP velocity becomes comparable to the velocity of the surrounding flow, causing Equation 4 to break down. However, the NP diffusion coefficient scales as the inverse of the number of linkers (S. Block, V. P. Zhdanov & F. Höök. Nano Letters 2016, 16, 4382), which means that the size range that can be investigated can in principle be tuned by varying the number of linkers.

Based on the suggestions of the reviewers, we decided to add a Supporting Section discussing the limitations of the approach more in detail (Supporting Section 10). There it is explained that the mobility of small NPs ($R \ll 200 \text{ nm}$) is given by the linker mobility, meaning that the NP movement is strongly decelerated by linking it to the SLB, such that the velocity of the surrounding fluid is much larger than the NP velocity. Hence, the NP velocity can be neglected in comparison to the fluid velocity, allowing us to express the acting hydrodynamic shear force via Equation 4 of the main text. Thus, for small NPs the hydrodynamic shear force is the read-out parameter that can be extracted from the data. This changes dramatically in the limit of large particles ($R \gg 200 \text{ nm}$). In this case, the

particle mobility is now dominated by friction coefficient of the particle within the solution, meaning that the SLB does not decelerate the particle movement anymore and the particle moves with the same velocity as the surrounding fluid (taken at the particle's midplane). This causes the hydrodynamic shear force acting on the particle to vanish (in comparison to a stationary particle of the same size) and hence, the particle size cannot be inferred from the hydrodynamic shear force anymore. However, as the particle moves with the same velocity as the surrounding fluid, its size should still be extractable from the measurement of its velocity, which is given by the product of shear rate times particle radius in this limit. Although we have not yet tested this modified data analysis to extend the applicable size range, it seems to be implementable, provided that the flow does not deform large particles. In addition, the transition between these 2 regimes can, as mentioned above, be extended toward larger R by linking the NPs using multiple linkers, as this increases the NP-bilayer frictional coefficient indicated by a decrease of the diffusion coefficient of the linked NPs.

ACTION: Following the suggestions of the reviewers, we have added Supporting Section 10 with discussion of the limitations of the approach in more detail.

I.7. Figure 2: Please define LMS.

ACTION: We revised the figure caption (marked in red) and avoided the acronym.

I.8. Figure 3: letters referring to the figure panels appear to be incorrect. Surely EM is a, b and flow nanometry is c, d?

ACTION: We thank the reviewer for making us aware of this typo and changed accordingly the caption for Figure 3.

Supporting information:

I.9. Section 1 duplicates a lot of the discussion in the main text and could be pruned accordingly.

ACTION: We agree with the reviewer and revised Supporting Section 1 (marked in red).

I.10. Figure 1e: solid line for average value is missing.

ACTION: We thank the reviewer and improved the optical appearance of the solid and dashed lines in the Supporting Figures 1d and 1e.

I.11. Supporting Movie 1: (i) What causes background of static (non-moving) particles? How are these particles removed from the analysis? (ii) While it is always dangerous to interpret such stochastic phenomena by eye, some of the particles appear to "stick" (i.e. stop diffusing) for short periods. Is this impression borne out by the tracking data, and what effect does it have on the results?

ANSWER:

(i) Such observations are usually attributed to imperfections of a SLB, which are always present at least to some extent. Nevertheless, we attribute non-moving gold NPs to be constituted by particles that are linked by more than 2 biotinylated lipids embedded within the SLB. This is motivated by the observation that (in contrast to gold NPs) vesicles do not show a notable immobile fraction after linking to the SLB, which rules out the possibility of sticking to bilayer imperfections, since the SLBs are formed in the same way in both types of experiments. Moreover, histograms of gold NP diffusion coefficients typically show 2 prominent peaks, which are (based on their position) attributed to linking via one and two biotinylated lipids (see Figure A1). The absence of higher order peaks, however, suggested that the gold NPs become immobile when linked to more than two biotinylated lipids, a behavior that is not yet fully understood. Non-moving gold NPs exhibit an apparent diffusion coefficient below $0.01 \mu\text{m}^2/\text{s}$, and are easily removed by considering only those NPs showing diffusion coefficients above $0.15 \mu\text{m}^2/\text{s}$. The absence of a peak in the diffusion coefficient histogram around zero (Figure A1) indicates that only a very small fraction of the overall data is contributed by non-moving gold NPs, caused by the fact that they remain fixed in the field of view while a much larger number of moving gold NPs pass through this area during the entire experiment.

(ii) We agree with the reviewer that some of the gold NPs show transient sticking of short duration. Under our experimental conditions, sticking will lead to an underestimation of the extracted D_x , D_y and v_x values, i.e., if D_{x0} , D_{y0} and v_{x0} denote the true diffusion coefficients and x-velocity of a mobile NP and p the mobile fraction of the NP trajectory ($p = 1$ refers to absence of sticking and $p = 0$ to sticking over the entire trajectory), the data analysis will yield pD_{x0} , pD_{y0} and pv_{x0} as extracted diffusion coefficients and x-velocity. However, this does not affect the extracted hydrodynamic shear force, since Eq. 4 (main text) contains the ratio of diffusion coefficient and x-velocity, causing p to drop out and thereby removing sticking-induced underestimation (similar to point I.2).

Figure A1: Diffusion coefficient histograms of gold NPs (hydrodynamic radius 30 nm) typically show 2 prominent peaks, which are attributed to gold NPs linked to one ($0.95 \mu\text{m}^2/\text{s}$) or two ($0.4 \mu\text{m}^2/\text{s}$) biotinylated lipids.

Reviewer #2

This represents an interesting and novel approach to the simultaneous monitoring of size and fluorescent intensity of cargo/markers within nanoparticles, which if shown to be generally useful will be the nanoscale version of flow cytometry. The author(s) refer to this new technique as 2D flow nanometry. The paper will be of interest to researchers who investigate the size dependence of NP interactions with cells, their uptake in environment settings etc., particularly those researchers who use fluorescent cargo or fluorescent markers to monitor uptake.

It is based on the idea that nanoparticles (NPs) can be tethered to supported lipid bilayers (SLBs) on the bottom of microfluidic channels and that the motion of the tethered NPs in the direction of flow (x-direction, parallel to channel) is decoupled from the random walk motion perpendicular to the flow direction (y-direction). In the direction of flow the diffusion of the NP is determined by the linker-lipid interaction (i.e. it is slow, due to drag through the SLB). The opposite is true for the motion perpendicular to the flow direction- it is determined by the NP- solution interaction. Unlike nanoparticle tracking analysis (NTA), the tether keeps the NP in the focal plane, so that accurate fluorescent intensity measurements can be made on the individual NPs. The random motion perpendicular to the flow direction can be monitored by optical microscopy to determine the diffusion coefficient and thus the hydrodynamic radius of the NPs. The power of the technique was clearly demonstrated in the SUV experiments by showing that the expected $I \sim R^2$ dependence was observed with much improved fidelity compared to NTA.

The paper is not likely to be one of the five most significant papers published in the discipline this year, but it definitely stands out from others in its field. However, if it really becomes the nano-version of flow cytometry, it could in the future be a seminal paper in nanotechnology. The claims are novel, particularly the constraint of the NPs to the field of view (via the tether) so that accurate intensities can be obtained.

We sincerely thank the Reviewer for his/her positive opinion about our work and constructive comments and suggestions, which will be addressed below.

II.1. What is not yet known is whether it will be easy to extend the work to other types of NPs of interest (e.g SLBs themselves) and NPs where extensive aggregation can occur. All of the derivations here assume isolated NPs.

ANSWER: We agree with the reviewer that the possibility to extend our work toward more complex NP architectures would extend the applicability of the method even further. It should be

immediately applicable to vesicle-like biological NPs (e.g., enveloped virions, micelles, exosomes, liposomes etc.), as the surface of these NPs is formed by lipid bilayers, which allows to link them via cholesterol-equipped DNA strands as presented in this work. Note that cholesterol also enters into gel phase bilayers, i.e., the lipid nanoparticle shell does not have to be in fluid phase to allow this linking procedure to be applied, making the approach very versatile. For NPs lacking a lipid shell one will have to find a suitable NP surface modification to enable linking to the SLB. We do not see this as a severe limitation, though, since a large number of linking strategies have been developed in the past decades, triggered by the interest to characterize nanoparticles by surface sensitive techniques.

We further agree with the reviewer that the approach was developed to characterize isolated NPs and that an extension towards aggregated NPs is not obvious, which are also difficult to characterize using complementary approaches. The shape of aggregated NP complexes usually deviates from a sphere, which is a key assumption in NTA, DLS and 2D flow nanometry and hence all these techniques are expected to report a kind of averaged hydrodynamic radius, i.e., a value between the smallest and largest extension of the NP complex. Due to the high accuracy of 2D flow nanometry in the determination of NP size and emitted intensity, there might in fact be the possibility to identify NP complexes by deviations for the expected size-intensity relation. This will still not allow the shape of the NP complex to be scrutinized (similar to NTA and DLS), but may allow characterizing the fraction of aggregated and non-aggregated NPs. Further, due to higher diffusion rates of NPs in bulk, the aggregation occurs faster for suspended than for tethered NPs. The method may thus aid the analysis of NP aggregation kinetics.

ACTION: Based on the suggestions of reviewer 1 and 2 (see also point I.6) we added a new section to the supporting information (page 17) to describe and discuss limitations in detail. We thank both reviewers for this suggestion.

II.2. For a nonspecialist, there are only two equations used:

$$D/(kBT) = vx/Fs, \text{ and } Fs(R) = A\eta v_0 R(R + \lambda)$$

where A is a constant prefactor (accounting for inhomogeneous flow around NP, which is fit), v_0 is the (known) flow rate through the channel and λ is the linker length (which is fit), η is the solution viscosity, and R is the radius of the NP. It could be made clearer how the actual sizes are obtained- i.e. particles of known R are used to calibrate the channel. What parameters are obtained from this calibration [(A and λ , and Fs from $D/(kBT) = vx/Fs$ (and vx from the linear increase in the x position of the NP)] that are then used for an unknown. It seems similar to calibrating GPC (SEC) columns, but this could be clearer. Maybe this information could be included as a supplementary method.

ACTION: Based on the suggestions of reviewer 2 and 3 (see also point III.2) we added a new section to the supporting information (page 12) to describe and discuss the calibration procedure in detail. We thank both reviewers for this suggestion.

II.3. Although this is an interesting technique to simultaneously monitor the size and intensity of single NPs, there is some clarification that would make the article better:

The author(s) should be clearer about whether the lipid bilayers need to be in the fluid phase for the method to work and how important and/or difficult it is to form a perfect lipid bilayer on the channel bottom.

ANSWER: The supported lipid bilayer, formed at the bottom of the channel, has to be in fluid phase and of high quality, to enable flow-induced NP movement after linking, which is a requirement for the applicability of the method. As pointed out above (point II.1), this does not apply for the bilayer of vesicle-like NPs, which can be in gel phase. The SLB has to be of high quality, i.e., exhibit few defects, as vesicle-like NPs tend to fuse with the SLB at defects, as indicated by transfer of dye-labeled lipids from the NP into the SLB. This process is easy to resolve in the microscopy movies and allows for visual assessment of the SLB quality. It is possible to reproducibly form SLBs of such quality, as judged from both our and many other groups' experience collected during the past years. We noticed that the addition of a small fraction of PEG-ylated lipids improves both fluidity and quality of the formed SLB and simplifies SLB formation.

ACTION: Based on the suggestions of reviewers 1 and 2 (see also point I.6) we added a new section to the supporting information (page 17) to describe and discuss limitations in detail. We thank both reviewers for this suggestion.

II.4. In addition, it was not obvious whether the gold NPs were used to calibrate the channels for the SUV experiments reported, and in general whether the same type of NP and linkers need to be used if different (unknown NPs) were being investigated.

ANSWER: We used indeed the gold NP data to calibrate the unknown parameters in Equation 4. We agree that we did not explicitly mention limitations regarding types of NP and linkers, which is now done in the new Supporting Section 10. In brief, since the NPs and linkers do not modify the flow profile at the bottom of the channel, the calibration is expected to be valid for all NPs that can be linked to the SLB.

ACTION: We thank the reviewer for the suggestions, which were incorporated in the new Supporting Section 10.

II.5. This was confusing for the SUV discussion, since the author(s) reported different distributions for measurements of SUVs using NTA, DLS and their method, without indicating how the SUVs were prepared.

ANSWER: All direct comparisons between NTA, DLS and 2D flow nanometry were done using vesicles from one and the same batch, i.e., in cases when size distributions obtained by different methods are compared in a single plot then these distributions were obtained by applying the indicated methods to vesicles originating from one and the same batch.

ACTION: We thank the reviewer for bringing this to our attention. We now indicated the preparation method of the vesicle batches shown in the Figures 4 and 5 and added more data in Supporting Section 7.

II.6. For their extruded SUVs, the pore size of the polycarbonate filter was not mentioned.

ACTION: Again, we thank the reviewer for bringing this to our attention and revised the corresponding part in the Materials and Methods section.

II.7. It is well known that sonication can produce different size distribution widths, and that this can also be true for extrusion, depending on the number of passes and polycarbonate pore size filter used (a minor detail is that vesicles within vesicles and double bilayers can also form).

ANSWER: We agree and note that this was the intention to use different preparation methods, i.e., we wanted to create vesicle batches that differed in their size distribution (range and polydispersity) in order to assess the performance of 2D flow nanometry. Significant presence of multilamellar vesicles within vesicles can be excluded as they would cause substantial deviations from the intensity-size relationship.

II.8. A comment about whether the SUVs could be deformed by the flow would be useful. If so, indicate whether this could affect the size analysis of the SUVs.

ANSWER: Flow-induced vesicle deformation would modify the extracted F_s and size distribution in dependence of the flow rate, i.e., different size distributions would be observed for different flow rate. This was, however, not observed in our experiments for vesicle diameter up to 150 nm,

suggesting that the applied hydrodynamic force is too small to cause membrane bending. Nevertheless, vesicle deformation can be induced by increasing the vesicle-SLB interaction, for example by linking the vesicles using charged lipids (e.g., positively charged lipids in the vesicles versus a negatively charged SLB), a behavior which we currently investigate in an independent set of experiments.

ACTION: We thank reviewer 2 and 3 (see also point III.7) for the suggestions, which were incorporated in the new Supporting Section 9. Together with the experimental arguments, we use there the theory and complementary experimental data in order to explain that the vesicle deformation is negligible in our case.

II.9. If there are limitations to the technique, a few sentences on this aspect would be informative. In particular, how will NPs that aggregate be handled?

ACTION: Based on the suggestions of reviewer 1 and 2 (see also point I.6) we added a new section to the supporting information (page 17) to describe and discuss limitations in detail. Again, we thank both reviewers for this suggestion.

Reviewer #3

The authors introduced a 2D flow cytometry device for separating nanoparticles. This reviewer cannot recommend the publication of this manuscript in Nature Communications due to lack of novelty and several technical issues.

We are surprised by this assessment by the reviewer, which is difficult for us to follow, since we are not aware of any related approach that shares the concepts presented in this work or any related study allowing to simultaneously quantify size and emitted intensity of single nanoparticles (radius < 50 nm). Already this capability is in our view a novel complement to existing methods. We would highly appreciate if the reviewer could supply references, which may help us to further differentiate our work from complementary approaches and stress the novel aspects. Further, although the concept opens up the possibility to separate (i.e., to sort) nanoparticles based on their individual properties (size and emitted intensity), a demonstration of this feature is considered beyond the scope of the present work.

III.1. Authors state that flow cytometry is incapable of fast detection of weak scattering of fluorescence signals to quantify the biological nanoparticles (BNPs). But this statement is not well

supported by references. Actually, many different flow cytometry methods have been reported over the years for BNP measurements.

ANSWER: We agree that this claim could have been better supported by references, which is now done in the revised version. It is true that flow cytometry has seen a tremendous development over the past years, but recent work on the characterization of extracellular vesicles clearly shows that conventional flow cytometers are only applicable to vesicles larger than 300 nm in diameter (see Refs. 12 and 13 in the main text), which can be improved by home-made modifications of the flow cytometer and labeling the vesicles using fluorescent dyes, yielding a resolution limit of 100 nm in diameter. The parameter plots shown in Figure 5 in the main text, however, clearly indicate that vesicles as small as 30 nm in diameter can be resolved by 2D flow nanometry, which is one order of magnitude better than reported for commercial flow cytometers (this point has been indicated on line 1 on page 12 in the main text). In addition, 2D flow nanometry allows to accurately quantify both the size and the intensity emitted by these small NPs, which we see, in agreement with the assessments of Reviewers 1 and 2, as a significant asset of our work.

ACTION: We thank the reviewer for bringing this issue to our attention and we supplemented the claim on page 3 with additional references.

III.2. Figure 2e. Only three sizes of gold nanoparticles were used in this calibration. Due to large error bars, this calibration is not reliable. Gold nanoparticles of different sizes (e.g., 60 and 80 nm) should also be measured.

ANSWER: We agree that it would be beneficial for the calibration procedure to rely on 5 instead of 3 sizes, but we disagree that calibration as presented in the first version of the manuscript was not reliable. The error bars given in Figure 2e indicate the standard deviation of the extracted hydrodynamic force, which is via Equation 5 coupled to the polydispersity of the NP size distribution. Hence, the presumably large error bar is a consequence of the NP polydispersity and does not reflect the reproducibility of the results, which is indeed much better (see e.g. Supporting Figure 4). To still improve on this aspect, we challenged the reliability of the calibration procedure in two steps including new data, first by a direct comparison of the size distributions of gold NPs obtained using electron microscopy and 2D flow nanometry, which covers (due to NP polydispersity) also the region mentioned by the reviewer, and in a second step by comparing the size distributions of 3 vesicle batches covering a similar size range as the gold NPs. We observed good agreement of the extracted size distributions in shape and peak position, indicative for the validity of the employed calibration procedure. Finally, the consistency of the data analysis is shown by adding the vesicle data to the calibration plot (size determined using NTA and F_s via 2D flow nanometry), supplying more data points as suggested by the reviewer. All these additional results together with detailed discussion of the calibration procedure are presented in the new Supporting Section 7.

ACTION: We thank the reviewer for making us aware of this potential shortcoming and added a novel section and new experimental data to the supporting information (page 12) to discuss the calibration procedure in detail.

III.3. Page 13, line 292 and 297. The authors state that the size accuracy of {plus minus} 2 nm for the gold NP and {plus minus} 1.5 nm for vesicles were achieved using 2D flow nanometry. However, the histograms of the size distributions in Figure 3 are clearly broader than what was stated.

ANSWER: We consider our statement to be correct. As in item III.2, we may articulate here that the widths of these distributions is given by the polydispersity of the NP sample and not the measurement accuracy of the approach, i.e., the 1.5 and 2 nm refer to the accuracy of 2D flow nanometry to extract the size of an individual tracked NP. The broader width is indicative for a relatively broad size distribution of the gold NPs, which is confirmed by electron microscopy. Note that distributions of 2D flow nanometry and electron microscopy are almost identical in shape, width and position, indicating that both approaches have similar accuracies in the size determination (since otherwise the width of the distribution would be systematically larger in one of the approaches).

ACTION: We thank the reviewer for bringing this to our attention and we revised the caption of Figure 3 accordingly (marked in red).

III.4. In Figure 4b and 4c, the band width of 2D flow nanometry is actually larger than that of NTA and DLS.

ANSWER: We agree. In the case of DLS (Figure 4c) we inadvertently plotted a wrong size distribution, which is corrected now. We thank the reviewer for making us aware of this error. Interestingly, we generally observed broader size distributions using 2D flow nanometry in comparison to NTA (see Supporting Figures 7 and 8), which is caused by a lack of small vesicles in the NTA distribution, while excellent agreement was observed for larger vesicle sizes. As DLS confirmed the presence of the small vesicle fractions, it seems that the NTA distributions were narrower because a part of the size distribution (the smaller vesicles) was not resolved by that method. This interpretation is supported by the comparison of the size distribution obtained by 2D flow nanometry and EM of gold NPs, in which case the histograms matched in both the small and large size regimes.

ACTION: We added further comparisons of NTA, DLS and 2D flow nanometry size distributions to the supporting information (page 14) and discussed accordingly in Supporting Section 7.

III.5. SI Movie 1 "95337_0_video_1733796_p6pr1x.avi". (i) How could the authors determine whether a moving particle is bound to the surface receptors in the supported lipid bilayer? Non-specific interactions can be another reason for the nanoparticles to stay on the lipid bilayer. (ii) Furthermore, if the particles bound to supported lipid bilayer move with the applied shear flow force, how to explain those particles that are not moving? (iii) Many of those stationary particles have strong signal, which may suggest that these particles are relatively large in size.

ANSWER:

(i) Specificity of the linking strategy is only important if the NP sample contains a certain sub-population that should be characterized using 2D flow nanometry (e.g., CD63-positive exosomes). This is here not the case, since entire NP samples have been characterized to assess the performance of the approach. Note that the nature of the linker is not important for 2D flow nanometry to work (as stated on page 8 in the main text and point I.2 in this document), but rather to achieve a linkage that allows for a 2D movement of the NP.

(ii) Non-moving NPs have so far only been observed by linking via streptavidin-biotin bonds and, although we do not know for sure the reason for this to occur, the analysis of the diffusion coefficient (see point I.11 in this document) suggests that non-moving NPs formed too many bonds to biotinylated lipids.

(iii) The gold NP measurements have been done using so-called surface enhanced ellipsometric contrast (SEEC) surfaces allowing a label-free imaging of the NPs (A. Gunnarsson, M. Bally, P. Jönsson, N. Médard & F. Höök. *Analytical chemistry* 2012, 84, 6538). As this approach monitors changes in the reflected intensity due to changes of the polarization plane and not scattering itself, the intensity-size relationship is non-trivial in this case, i.e., brighter signals are not necessarily caused by larger NPs. Predominant immobilization of larger NPs would be reflected by a bias of the extracted size distributions toward smaller NP sizes, which is not observed in our experiments.

Other comments:

III.6. Page 4, line 64 to 67: The BNPs linked to SLB still performing Brownian movement perpendicular to the flow direction. It should be interesting to quantify the effects of hydrodynamic forces on the magnitude of the Brownian motion.

ANSWER: We show in Supporting Section 3 that the 2D trajectories can be decomposed into a 1D directed movement (caused by the hydrodynamic shear force) and a 2D Brownian motion, having equal diffusion coefficients in direction of and perpendicular to the flow (see Supporting Figure 2a).

This indicates that the flow does not affect the Brownian motion, i.e., the random component of the movement is not modified by the presence of a hydrodynamic shear force, which is also reflected by absence of flow-induced modifications of the diffusion coefficient distributions. Here, we may add that the absence of the effect of a particle drift on its diffusion is implied in all the related theoretical works since the seminal articles by Einstein and Smoluchowski who introduced relation (4).

III.7. Page 11, line 235 and 236: How will the morphology of the vesicles change in response to the hydrodynamic shear flow forces?

ANSWER: This interesting question was raised also by reviewer 2 and is addressed in point II.8 in this document. In order to avoid duplications, we refer to page 10 above and the new Supporting Section 9 for the answers.

ACTION: We thank reviewer 2 and 3 (see also point II.8) for the suggestions, which were incorporated in the novel Supporting Section 9.

III.8. In Page 8, line 170 and Page 9, line 173, the authors stated that length λ is negligible. However, in Page 14, line 311 and 312, they make a different point: "Note that the length λ in Eq. 5, (connecting R and Fs) was found to be important, since attempts to fit Eq. 5 failed for $\lambda = 0$."

ANSWER: We agree that this is a confusing statement. The correct former statement is that the influence of the linker length on λ is negligible, which follows from experiments using the same gold NP batch but different linking strategies and which yielded the same distribution of the hydrodynamic force (see Supporting Figure 4). In theory, one would also expect λ to be negligible, as the so-called slip length is usually reported to be on the order of a few nm. However, initial attempts to fit Eq. 5 using $\lambda = 0$ failed to properly describe the calibration measurements, which made it necessary to introduce λ as fitting parameter, as commonly done also by others (E. Bonaccorso, M. Kappl & H. J. Butt. Physical Review Letters 2002, 88, 076103). We see this as a strength of this work, as the relationship between size and hydrodynamic force is extracted from dedicated experiments.

ACTION: We thank the reviewer for bringing this to our attention and rephrased on page 9 accordingly (marked in red).

III.9. for Figure 3 (Page 22) is incorrect. Figure 3a and 3b are electron microscopy measurements, while Figure 3c and 3d are 2D flow nanometry measurements.

ANSWER: We thank the reviewer for making us aware of this typo.

ACTION: We changed accordingly in the caption of Figure 3.

III.10. The references have inconsistent formats.

ACTION: We thank the reviewer for bringing this to our attention and revised the references.

REVIEWERS' COMMENTS:

Reviewer #1 (Remarks to the Author):

I was very pleased to see the substantial revisions made to this interesting manuscript in response to my comments, and those of the other reviewers. The authors' responses are constructive, comprehensive, and clear and address all of my questions and concerns. I would be very happy to see this manuscript accepted and I think it will be influential in the coming years.

Reviewer #2 (Remarks to the Author):

The authors have more than adequately answered any questions and have made the manuscript very clear.

We sincerely thank the Reviewers for their positive opinion about our work and their constructive comments and suggestions of the first review.